# Neutralizing Antibodies in COVID-19 Serum from Tatarstan, Russia

**DOI:** 10.3390/ijms241210181

**Published:** 2023-06-15

**Authors:** Shaimaa Hamza, Ekaterina Martynova, Ekaterina Garanina, Venera Shakirova, Alisa Bilalova, Svetlana Moiseeva, Ilsiyar Khaertynova, Olesia Ohlopkova, Nataliya Blatt, Maria Markelova, Svetlana Khaiboullina

**Affiliations:** 1OpenLab “Gene and Cell Technologies”, Kazan Federal University, 420021 Kazan, Russia; shaimaa.hamza@mail.ru (S.H.); ignietferro.venivedivici@gmail.com (E.M.); ekaterinaakagaranina@gmail.com (E.G.); nataliya.blatt@gmail.com (N.B.); mimarkelova@gmail.com (M.M.); 2Department of Infectious Diseases, Kazan State Medical Academy, 420012 Kazan, Russia; vene-shakirova@yandex.ru (V.S.); alisa-bilalova@mail.ru (A.B.); sgerasimova.kgma@gmail.com (S.M.); i.khaertynova@gmail.com (I.K.); 3State Research Center of Virology and Biotechnology «Vector» of Rospotrebnadzor, 630559 Koltsovo, Russia; ohlopkova.lesia@yandex.ru

**Keywords:** neutralizing antibodies, COVID-19, serum, Th1 lymphocyte

## Abstract

The severity of COVID-19 is a result of the complex interplay between various branches of the immune system. However, our understanding of the role of neutralizing antibodies and the activation of cellular immune response in COVID-19 pathogenesis remains limited. In this study, we investigated neutralizing antibodies in patients with mild, moderate, and severe COVID-19, analyzing their cross-reactivity with the Wuhan and Omicron variants. We also assessed the activation of the immune response by measuring serum cytokines in patients with mild, moderate, and severe COVID-19. Our findings suggest the early activation of neutralizing antibodies in moderate COVID-19 compared to mild cases. We also observed a strong correlation between the cross-reactivity of neutralizing antibodies to the Omicron and Wuhan variants and the severity of the disease. In addition, we found that Th1 lymphocyte activation was present in mild and moderate cases, while inflammasomes and Th17 lymphocytes were activated in severe COVID-19. In conclusion, our data indicate that the early activation of neutralizing antibodies is evident in moderate COVID-19, and there is a strong correlation between the cross-reactivity of neutralizing antibodies and the severity of the disease. Our findings suggest that the Th1 immune response may play a protective role, while inflammasome and Th17 activation may be involved in severe COVID-19.

## 1. Introduction

The COVID-19 pandemic, declared by the World Health Organization (WHO) on 11 March 2020, is ongoing, with over 541 million cases diagnosed worldwide [1]. COVID-19 is a disease characterized by symptoms such as fever, cough, sore throat, shortness of breath, loss of sense of smell and taste [2,3], and in severe cases, lower respiratory tract infection [3]. The disease can range from asymptomatic to mild, moderate, severe, or critical forms [3], and there is a correlation between age and comorbidities and the severity of the disease. Older age and underlying medical conditions increase the likelihood of severe and critical forms of COVID-19 [4,5].

One of the factors contributing to the severity of the disease is the absence of neutralizing antibodies. Neutralizing antibodies have been detected in some patients as early as the first week after symptom onset [6], and the number of patients with circulating neutralizing antibodies increases as the disease progresses. The majority of these antibodies are directed against the spike (S) protein [6,7], which mediates SARS-CoV-2 cell entry by binding to the angiotensin-converting enzyme 2 receptor on the host cell membrane [8]. Monitoring of neutralizing antibodies has been shown to have predictive value for critical and fatal COVID-19. It has been demonstrated that the lack of circulating neutralizing antibodies in the early stage of the disease could be a marker of a fatal outcome [6], and the level of neutralizing antibodies increases with disease severity [9,10]. These findings support the development of neutralizing antibodies during COVID-19 and their potential use in predicting the critical form of the disease.

There have been several strains of SARS-CoV-2 that have emerged during the pandemic, including Alpha (B.1.1.7 and Q lineages), Beta (B.1.351 lineage), Gamma (P.1 lineage), Delta (B.1.617.2 and AY lineages), Epsilon (B.1.427 and B.1.429 lineages), and others [11]. Some of these strains, such as B.1.1.7, BA.1, BA.1.1, and BA.2, have been shown to be highly contagious [11,12,13], while others, such as B.1.617.2, B.1.17, and B.1.351, are associated with a higher risk of severe COVID-19 [14,15]. Several studies have demonstrated that neutralizing antibodies produced during infection with one lineage of SARS-CoV-2 may have cross-reactivity and efficacy against other lineages [16,17]. This is especially important since many vaccines currently available were made using the original Wuhan strain of SARS-CoV-2 [18,19,20]. By producing cross-reactive neutralizing antibodies, protection may be maintained even when infection is caused by a different variant of the coronavirus.

SARS-CoV-2 infection was first documented in Russia in March 2020, where cases were linked to travel abroad and local transmission [21]. Initially, COVID-19 was associated with the Wuhan strain of SARS-CoV-2 infection, but later, the AY.122 sub-lineage of the Delta variant was frequently identified in collected samples [22]. In November 2021, a new variant of the SARS-CoV-2 virus, named Omicron, was first reported in Russia [23]. By 3 January 2022, Omicron had been detected in approximately 50% of COVID-19 cases. By February 2022, the Omicron variant of SARS-CoV-2 was found to be responsible for 100% of the reported COVID-19 cases. SARS-CoV-2 can elicit an antibody immune response in infected patients that can last for up to 6 months [24]. Interestingly, acute and convalescent serum samples were shown to react with multiple peptides located in the receptor-binding region (RBD) of the S protein [25]. This observation suggests that some of these antibodies may have neutralizing activity, as the RBD region of the SARS-CoV-2 S protein has been previously identified as the target of neutralizing antibodies [26]. Additionally, data reveal that neutralizing activity correlates with anti-RBD antibody levels [27]. However, the presence of neutralizing antibodies in the serum of COVID-19 patients in Russia remains largely unknown, as does their cross-reactivity and role in the development of the various forms of the disease.

COVID-19 was first diagnosed in the Republic of Tatarstan, Russia, in March 2020 [28]. Since then, several variants of the SARS-CoV-2 virus, such as Delta and Omicron, have been isolated from COVID-19 patients in Russia [29]. While there are limited data available on the antibody response to SARS-CoV-2 [30], even less is known about the production of neutralizing antibodies in COVID-19 patients in Russia [31]. To address this knowledge gap, our study aimed to analyze the humoral immune response in mild, moderate, and severe COVID-19 patients from the Republic of Tatarstan, Russia, with a specific focus on studying neutralizing antibodies. In addition, we studied the cross-reactivity of neutralizing antibodies in COVID-19 patients. Additionally, we assessed serum cytokines in mild, moderate, and severe COVID-19 patients.

## 2. Results

### 2.1. Clinical Characteristics of COVID-19 Patients

Patients were classified based on the severity of COVID-19 as mild, moderate, and severe. We found no significant difference in age and sex between patients with mild, moderate, and severe forms of COVID-19. However, there was a higher death rate (51.9%) in patients with severe COVID-19 compared to those with mild (0%) and moderate (3.33%) COVID-19. Additionally, there were higher frequencies of patients receiving SGCT and olokizumab in patients with moderate and severe COVID-19.

Initially, clinical laboratory data were compared between each form of COVID-19 and the controls (Table 1). We found that all forms of COVID-19 patients had increased monocyte counts and TT as well as increased serum levels of CRP, fibrinogen, and total anti-SARS-CoV-2 antibodies compared to the controls. There were also severity-specific differences depending on the severity of COVID-19. In the mild form, patients had increased neutralizing antibodies to the SARS-CoV-2 Wuhan strain. Increased serum levels of neutralizing antibodies to SARS-CoV-2 Omicron and Wuhan were found in moderate COVID-19. In contrast, limited changes in serum neutralizing antibodies to either SARS-CoV-2 strain were found in severe COVID-19. Instead, the severe form of COVID-19 had an increased serum level of AST and decreased leukocyte counts.

Next, we compared clinical laboratory data between patients with different COVID-19 severity (Table 1). We found a higher CT%, as well as serum levels of CRP, neutralizing antibodies to the SARS-CoV-2 Omicron strain, and total anti-SARS-CoV-2 antibodies in moderate compared to mild forms of COVID-19. A higher CT%, leukocyte count, and serum levels of CRP and AST were found in the severe compared to mild forms of COVID-19. In addition, lower monocyte and lymphocyte counts were found in severe compared to mild COVID-19. Only the CT% and serum CRP were higher in severe compared to moderate COVID-19. Additionally, the monocyte and lymphocyte counts as well as anti-SARS-CoV-2 neutralizing antibodies to Omicron and total SARS-CoV-2 antibodies were lower in severe compared to moderate COVID-19.

### 2.2. Neutralizing Antibodies to Omicron and Wuhan in Serum of COVID-19 Patients 

During the time of COVID-19 serum collection (January–March 2022) [32], the SARS-CoV-2 Omicron variant was prevalent in Russia. Therefore, we tested all 81 samples for the presence of anti-SARS-CoV-2 Omicron variant neutralizing antibodies (Table 2). We found that the highest number of patients positive for anti-Omicron variant (18 out of 30; 60%) neutralizing antibodies were in the moderate COVID-19 group. Patients with mild and severe forms had anti-Omicron variant neutralizing antibodies in 6 out of 24 (25%) and 5 out of 27 (18.5%) samples, respectively.

We observed that patients with moderate COVID-19 exhibited higher levels of Omicron neutralizing antibodies compared to those with mild and severe COVID-19 (Table 2, Figure 1). However, we did not find significant differences in the inhibition rate of neutralizing antibodies between fatal and non-fatal COVID-19 cases, patients who received olokizumab therapy and those who did not, or among individuals of different ethnicities (Appendix A). Interestingly, patients who did not undergo standard glucocorticoid therapy (SGCT) had higher levels of neutralizing antibodies against Omicron, primarily observed in the mild COVID-19 group (36.5 ± 39.0% versus 16.2 ± 30.8%). We did not find any differences in the frequency of neutralizing antibodies against the Wuhan and Omicron lineages of SARS-CoV-2 between patients who received olokizumab treatment and those who did not (Appendix A). Our findings indicate that the ethnicity of patients does not contribute to the severity of the disease, as there were no significant differences in the frequency of mild, moderate, and severe forms of COVID-19 among patients of various ethnic backgrounds (Appendix A). Additionally, no differences were observed in the frequency of neutralizing antibodies among patients of different ethnicities who had mild, moderate, and severe forms of the disease (Appendix A).

We also tested 50 COVID-19 serum samples for cross-reactive neutralizing antibodies to the Wuhan variant. We selected all 29 serum samples from moderate COVID-19 as this group had the highest number of patients positive for neutralizing antibodies to the Omicron variant. Additionally, we included 10 and 11 serum samples from mild and severe COVID-19, respectively. Only serum samples positive for anti-Omicron neutralizing antibodies from the mild and severe COVID-19 patients were selected for this study. We found 100% concurrence between positive and negative Omicron and Wuhan neutralizing antibodies in the mild COVID-19 serum samples tested (6 positive and 4 negative, both Omicron and Wuhan). With the exception of two serum samples, the neutralizing antibody results for the Omicron and Wuhan variants also showed a high degree of correlation. Two serum samples that were positive for Omicron were negative for Wuhan neutralizing antibodies. The correlation analysis of neutralizing antibodies to Omicron and Wuhan is shown in Figure 2, demonstrating a strong correlation between the detection of neutralizing antibodies to Omicron and Wuhan (R = 0.95, *p* < 0.00001).

### 2.3. Cytokine Activation in Mild, Moderate, and Severe COVID-19 Patients Compared to Controls 

The levels of 33 cytokines were found to be increased, while 5 were decreased in patients with mild COVID-19 (Figure 3). In contrast, a lower number of cytokines, 25 in total, were increased in patients with moderate disease. Additionally, the levels of six cytokines were lower than in the control group. Interestingly, 22 cytokines were activated in both mild and moderate COVID-19, while 4 were lower than in the control group. Notably, IL-12p40 and IL-12p70 were found to be higher only in mild COVID-19, while IL-12p40 was even lower in moderate COVID-19 compared to the control group.

In severe COVID-19, a total of 25 cytokines were activated, which is similar to the number found in moderate COVID-19. Additionally, five cytokines were lower than in the control group. Of the activated cytokines, 16 were also increased in mild and moderate COVID-19. However, some cytokines, including IL-1β, IL-3, IL-17, and CCL3, were uniquely activated in severe COVID-19.

### 2.4. Comparison Analysis of Cytokine Activation between Mild, Moderate, and Severe COVID-19 Patients

In moderate COVID-19, 3 cytokines had higher serum levels, while 16 cytokines had lower levels compared to mild cases (Figure 4). However, more significant changes in serum cytokine levels were observed between mild and severe COVID-19. Specifically, 9 cytokines had higher levels, and 20 cytokines had lower levels in severe cases compared to mild cases. In addition, seven cytokines had higher levels, and six cytokines had lower levels in severe cases compared to moderate cases of COVID-19.

## 3. Discussion

In this study, we analyzed neutralizing antibodies in the serum samples of COVID-19 patients who were hospitalized in the Agafonov Republican Clinical Hospital for Infectious Disease, Republic of Tatarstan, between January and March 2022. COVID-19 was first diagnosed in Kazan, Russia, in March 2020 [33], and it can manifest in mild, moderate, or severe forms. Severe COVID-19 is characterized by the progressive deterioration of the patient’s health, requiring hospitalization and admission to the ICU (critical COVID-19). Studies have shown that the humoral immune response is essential for protection from SARS-CoV-2 infection [34,35]. However, hyperactivation of the immune response and hyperinflammation could lead to tissue damage and multi-organ failure [36,37]. The outcome of the complex interplay between the humoral immune response and pro-inflammatory cytokine activation appears to manifest in the form of disease severity.

We found that the detection of anti-SARS-CoV-2 Omicron variant neutralizing antibodies varies in patients with different severities of COVID-19. The highest number of patients positive for Omicron neutralizing antibodies was in moderate COVID-19, while these antibodies were found in a smaller number of patients with mild and severe forms of COVID-19. 

It is important to note that during January 2022, both the Delta variant and the Omicron variant of SARS-CoV-2 were circulating in Russia simultaneously [23]. However, our study specifically focused on the detection of Omicron SARS-CoV-2 neutralizing antibodies in the COVID-19 patients included in this study. It is possible that some patients may have been infected with the Delta variant and developed non-cross-reacting neutralizing antibodies. Nevertheless, the key finding of our study is that neutralizing antibodies against the Omicron variant of SARS-CoV-2 were produced in COVID-19 patients. These antibodies are expected to provide protection against reinfection with the Omicron variant, which became the dominant strain in Russia after February 2022. Our data also support previous observations that neutralizing antibody detection is not associated with the mild form of the disease [38,39]. We also found that a lesser number of severe compared to moderate patients had Omicron neutralizing antibodies. This could be explained by the observation made by Lucas et al., demonstrating a delay in the development of neutralizing antibodies in severe COVID-19 patients [40]. Additionally, the lower prevalence of anti-SARS-CoV-2 antibodies among severe COVID-19 patients could be attributed to the observed decrease in CD19+ lymphocyte numbers in this particular patient group [41]. Papas et al. conducted a study that revealed significantly lower B cell counts in severe COVID-19 patients, and these counts did not recover throughout the course of the disease [41].

Our data suggest that the severity of the disease has limited correlation with the frequency of neutralizing antibody detection. Therefore, other factors, in addition to neutralizing antibodies, could contribute to the clinical form of the disease. One of these factors could be inflammation and the activation of the immune response. Supporting this assumption are our findings of the substantial changes in the quantity and quality of cytokines activated in mild, moderate, and severe COVID-19. We found more cytokines activated in mild compared to severe and moderate COVID-19. However, the striking observation was the upregulation of IL-12p40 and IL-12-p70 in mild but not in moderate and severe COVID-19. Another interesting finding was the increased serum levels of IL-1β, IL-3, IL-17, and CCL3 in severe COVID-19 exclusively.

IL-12 is essential for initiating the differentiation of naïve CD4+ T helper (Th) cells [42]. In addition, IL-12 stimulates the production of IFN-γ [43], a cytokine that promotes the differentiation of Th1 cells [44]. Another cytokine that stimulates IFN-γ production is IL-18 [45]. The current paradigm of Th1 cell differentiation states that IL-12 is an inducer, while IL-18 is an activator of Th1 lymphocytes [46]. The serum levels of IL-12, IL-18, and IFN-γ were increased in mild COVID-19 compared to the control group. However, only the levels of IL-18 and IFN-γ increased, while IL-12 remained unaffected in moderate COVID-19 compared to the control group. The serum levels of IL-12, IL-18, and IFN-γ differed even more in the severe compared to moderate and mild forms of COVID-19. The serum level of only IL-18 was increased, while IL-12 and IFN-γ appeared to be unaffected in severe COVID-19 when compared to the control group. In addition, IL-12 was lower than that in mild and moderate COVID-19. Another piece of evidence supporting the activation of Th1 lymphocytes in mild and moderate COVID-19 is the increased serum IL-13 level in these patients. In contrast, IL-13 levels were not affected in severe COVID-19 patients. IL-13 was shown to be produced by Th1 stimulated with IL-2, IL-18, and antigen [47]. Our data demonstrate that the serum levels of IL-12, IL-13, IL-18, and IFN-γ are disturbed in patients with severe COVID-19 compared to mild and moderate cases. The early activation of cytokines supporting the differentiation of Th1 cells, such as IL-12, IL-18, and IFN-γ, appears to contribute to the mild form of the disease. This assumption is supported by Gil-Etayo et al., who demonstrated that early Th1 immune response could produce favorable disease progression [48]. In addition, the potentially beneficial role of the Th1 immune response was reported in COVID-19 patients with asthma [49]. Therefore, it could be suggested that the severity of the disease could be the result of the efficacy of the Th1 response activation in COVID-19 patients.

Another notable finding was the elevated levels of IL-1β and IL-17 in the serum of severe COVID-19 patients compared to the control group. IL-1β is a pleiotropic cytokine that is produced by activated inflammasomes [50]. Boraschi summarizes the role of IL-1β in the pathogenesis of the disease as initiating and amplifying the inflammatory response, as well as recruiting and activating leukocytes [51]. IL-1β can also support T cell priming [52] and enhance the release of IFN-γ and IL-17 by CD4+ T cells [53,54]. On the other hand, IL-17 is primarily produced by Th17 lymphocytes and is implicated in the pathogenesis of immunopathology [55]. IL-17 can work synergistically with IL-1β and TNF-α to enhance inflammation [56]. Hence, the role of IL-17 in COVID-19 has been recognized as an “inflammation amplifier” [57].

We have found higher levels of total anti-SARS-CoV-2 antibodies and neutralizing antibodies in mild and moderate COVID-19 patients compared to severe cases during the early days of hospitalization. These findings support previous observations by Lucas et al., who reported a delayed anti-spike protein IgG response in fatal COVID-19 cases compared to survivors [40]. Ren et al. also reported the delayed onset of the humoral immune response in severe COVID-19 cases compared to mild and moderate cases [58]. Similarly, Zhang et al. showed a delayed peak of anti-SARS-CoV-2 antibodies [59], while mild and moderate COVID-19 cases had the highest antibody levels and reached them earlier, two to three weeks, after symptom onset [60]. These data suggest that the early analysis of anti-SARS-CoV-2 antibodies could be used as a predictor factor for fatal COVID-19.

One interesting observation was a strong correlation between the detection of neutralizing antibodies to Omicron and Wuhan. This suggests that neutralizing antibodies to Omicron have some degree of neutralizing activity against Wuhan. Previous studies have reported the presence of cross-reacting neutralizing antibodies to Omicron and other SARS-CoV-2 variants in convalescent COVID-19 serum [61,62], indicating a possibility of previous exposure to other SARS-CoV-2 variants. Park et al. suggested that Wuhan cross-reacting antibodies in Omicron COVID-19 could be the result of immunological imprinting [63], which is supported by the fact that neutralizing antibodies were found in the serum of acute, early stage COVID-19 cases. This immunological imprinting could also explain our finding of a higher level of neutralizing antibodies to Wuhan, not to Omicron, in mild COVID-19 cases compared to the control group. Only moderate COVID-19 serum had neutralizing activity against both Omicron and Wuhan. Our findings corroborate observations by da Silva et al., who demonstrated that serum from moderate COVID-19 cases had a higher neutralizing ability against many SARS-CoV-2 variants compared to mild or severe forms [64].

In conclusion, our study found differences in the total antibody and neutralizing antibody levels among different severity levels of COVID-19. Patients with moderate disease had higher levels of antibodies compared to those with mild or severe disease. Additionally, we observed cross-reactivity between neutralizing antibodies to the Omicron and Wuhan variants. However, only a smaller group of patients with mild and severe COVID-19 were positive for neutralizing antibodies. These findings suggest that the development of neutralizing antibodies alone may not be the sole determinant of disease severity.

Furthermore, we found that cytokine levels in the serum differed between severe and mild/moderate COVID-19. Mild and moderate COVID-19 patients had increased cytokines associated with the Th1 immune response, while severe COVID-19 patients had a cytokine signature, indicating the activation of Th17 lymphocytes and inflammation. Our results suggest that the delayed development of neutralizing antibodies, combined with severe inflammation due to inflammasome activation and Th17 immune response, may contribute to the severity of COVID-19.

## 4. Materials and Methods

### 4.1. Human Subjects

Acute serum samples were collected from 81 COVID-19 patients (32 males and 49 females) admitted to the Agafonov Republican Clinical Hospital for Infectious Disease in the Republic of Tatarstan between January and March 2022. The patients had an average age of 67.0 ± 14.7 years old. Clinical characteristics of COVID-19 were also collected for these patients and are presented in Table 1. SARS-CoV-2 infection was diagnosed based on clinical signs and symptoms and confirmed by detection of SARS-CoV-2 RNA using quantitative polymerase chain reaction (qPCR). Samples were collected on third day post-admission to the hospital. All serum samples were aliquoted and stored at −80 °C.

Additionally, serum samples from 26 age-matched controls (11 males and 15 females) with an average age of 62.8 ± 13.6 years old were collected. All control samples were tested negative for anti-SARS-CoV-2 antibodies. 

### 4.2. COVID-19 Treatment 

All patients received standard glucocorticoid therapy (SGCT) as part of their treatment. However, for patients who did not show signs of clinical improvement (such as increased body temperature, lower PaO_2_ and PaO_2_/FiO_2_, progressive respiratory insufficiency, etc.), an additional intervention was administered. These patients were given olokizumab, an IL-6 binding monoclonal antibody [65], 24 h after initiating SGCT. The SGCT regimen began upon admission and involved the intravenous administration of methylprednisolone (60 mg) every 6 h for a duration of 4 days. The dosage was subsequently reduced by 20–25% every 1–2 days, with a 50% decrease occurring after 2 days. Twenty-four hours after initiating SGCT, patients received a single dose of olokizumab (256 mg) via intravenous injection. A schematic presentation of the COVID-19 treatment approach is depicted in Figure 5.

### 4.3. Criteria for the Mild, Moderate, and Severe Forms of COVID-19

The severity of COVID-19 was determined following the recommendations published in the 17th edition of “Prophylaxis, Diagnosis, and Treatment of New Coronavirus Infection (COVID-19)” [66].

Mild COVID-19 signs and symptoms included a fever < 38 °C, no changes in breathing rate, a saturation pressure of oxygen (SpO_2_) ≥ 97%, and blood pressure (BP) within the normal range.

Moderate COVID-19 signs and symptoms were defined as a fever > 38 °C, a breathing rate > 22/min, an SpO_2_ < 95%, a ratio of arterial oxygen partial pressure to fractional inspired oxygen (PaO_2_/FiO_2_) ≤ 300 mmHg, blood pressure (BP) within the normal range, the presence of signs of lung damage in CT, and shortness of breath during physical exertion.

The severe form of COVID-19 was diagnosed based on the following signs and symptoms: a fever > 38 °C, a breathing rate > 30/min, an SpO_2_ ≤ 93%, a PaO_2_/FiO_2_ ≤ 300 mmHg, agitation, decreased consciousness, BP < 90/60 mmHg, the presence of signs of lung damage in CT, and a quick Sequential Organ Failure Assessment (qSOFA) score > 2 units.

### 4.4. Ethics Statement

The ethics committee of the Kazan Federal University approved this study, and signed informed consent was obtained from each patient and control subject according to the guidelines adopted under this protocol (Protocol 4/11 of the KMSA Ethics Committee meeting dated 10 November 2020). Institutional Review Board of the Kazan Federal University, and informed consent was obtained from each respective subject according to the guidelines approved under this protocol (Article 20, Federal Law of Health Right of Citizens of Russian; N323-FZ, 21 November 2011).

### 4.5. Anti-SARS-CoV-2 Antibody Enzyme-Linked Immunosorbent Assay (ELISA)

The SARS-CoV-2-CoronaPass ELISA kit (Genetico, Moscow, Russia) was used to detect pan-SARS-CoV-2-specific antibodies according to the manufacturer’s instructions. Briefly, COVID-19 and control sera were mixed with conjugate-1 at a 1:10 ratio and incubated for 30 min at 37 °C in a 96-well plate pre-adsorbed with SARS-CoV-2 antigens. Inactivated human serum without SARS-CoV-2 antibodies was used as a negative control (provided within the kit). Following washes (3×; 0.5% Tween20 in PBS, PBS-T), wells were incubated with anti-human IgG + IgM + IgA horse radish peroxidase (HRP) conjugated antibodies for 30 min at 37 °C. After washing the wells 3 times with 0.5% Tween20 in PBS, they were incubated with 3,3’,5,5’ Tetramethylbenzidine (TMB) (Chema Medica, Moscow, Russia). Data were acquired using a Tecan 200 microplate reader (Tecan, Zürich, Switzerland) at OD450 with a reference OD650. A positive result was recorded when the ratio of the tested sample OD450 to the negative control OD450 + 0.15 was greater than 1.

### 4.6. Neutralizing Assay

The SARS-CoV-2 Surrogate Virus Neutralization Test Kit (GenScript, Nanjing, China) was used to analyze neutralizing antibodies against SARS-CoV-2 Omicron and Wuhan types individually by following the manufacturer’s instructions. This test was clinically validated by the manufacturer using the Plaque Reduction Neutralization Assay (PRNT) as a standard assay to detect neutralizing antibodies [67]. The manufacturer states that the GenScript cPass SARS-CoV-2 neutralizing test has 100% positive agreement with PRNT_50_ (95% CI 87.1–100.0%) and 100% positive agreement with PRNT_90_ (95% CI 87.1–100.0%). In brief, HRP-receptor-binding domain (RBD) of SARS-CoV-2 spike protein (1:1000) was added to each positive control, negative controls (provided by manufacturer), and patient serum sample at a 1:1 ratio and incubated at 37 °C for 30 min. Patient serum samples were diluted using phosphate saline buffer (PBS; pH 7.2). Then, 100 μL of the positive control, negative control, and sample mixed with HRP-RBD was added to wells pre-coated with hACE2 protein and incubated at 37 °C for 15 min. Plates were washed 4 times with 260 μL of 1× wash solution (provided by the manufacturer) and 100 μL of TMB was added to each well and incubated in the dark at 20–25 °C for 15 min. Stop solution (50 μL) was added before data were collected using a Tecan 200 microplate reader (Tecan, Switzerland) at OD450 nm. The inhibition rate % was calculated from the equation: inhibition rate = [1 − (OD value of Sample/OD value of Negative control)] ∗ 100%. 

Samples with inhibition rate ≥30% of the cut-off value were identified as positive for SARS-CoV-2 neutralizing antibodies. Samples with inhibition rate values below 30% of the cut-off were classified as negative for SARS-CoV-2 neutralizing antibodies.

### 4.7. Multiplex Analysis

The Bio-Plex Pro Human Cytokine 48-plex Screening Panel (12007283, BioRad, Hercules, CA, USA) was utilized to measure serum cytokines. The serum cytokine levels were determined using the Bio-Plex (Bio-Rad, Hercules, CA, USA) multiplex magnetic bead-based antibody detection kit following the manufacturer’s instructions. Serum samples (50 μL) were analyzed with at least 50 beads per analyte. Serum samples were mixed with pre-mixed cocktail of antibody-coated magnetic beads (50 μL). Washed (3x, wash buffer), antibody–beads complexes were incubated with pre-mixed cocktail of biotinylated detection antibody. Washed (3×; wash buffer) antibody–beads complex was incubated with Streptavidin-PE. Mass-calibrated standards were provided by manufacturer and used to generate calibration curve. Median fluorescence intensities were measured using a MAGPIX analyzer (Luminex, Austin, TX, USA), and each sample was evaluated in triplicate. The data collected were analyzed using MasterPlex CT control software and MasterPlex QT analysis software 512 (MiraiBio, San Bruno, CA, USA). Standard curves were generated for each cytokine using the standards provided by the manufacturer.

### 4.8. Statistical Analysis

Statistical analysis was conducted using the R environment [67]. A *p*-value < 0.05 was considered statistically significant and was determined using the Kruskal–Wallis test with Benjamini–Hochberg adjustment for multiple comparisons. The distribution of sex and death rates, as well as the frequency of patients receiving standard glucocorticoid therapy (SGCT) and olokizumab in COVID-19 and control groups, were assessed using Fisher’s exact test. Correlations were analyzed using the R psych package, based on Spearman’s rank correlation coefficient. *p*-values were adjusted with the Benjamini–Hochberg method.

## Figures and Tables

**Figure 1 ijms-24-10181-f001:**
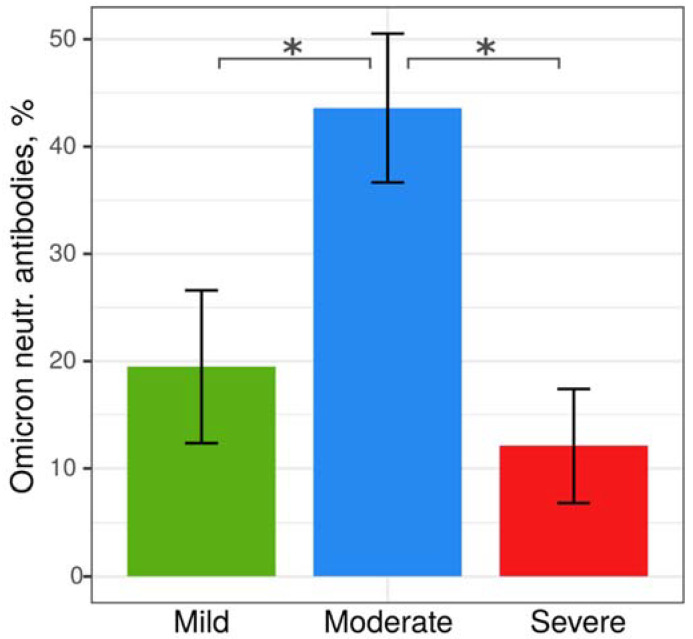
Anti-SARS-CoV-2 Omicron variant neutralizing antibodies in patients with different severity of COVID-19 (* presented as mean ± standard error of mean).

**Figure 2 ijms-24-10181-f002:**
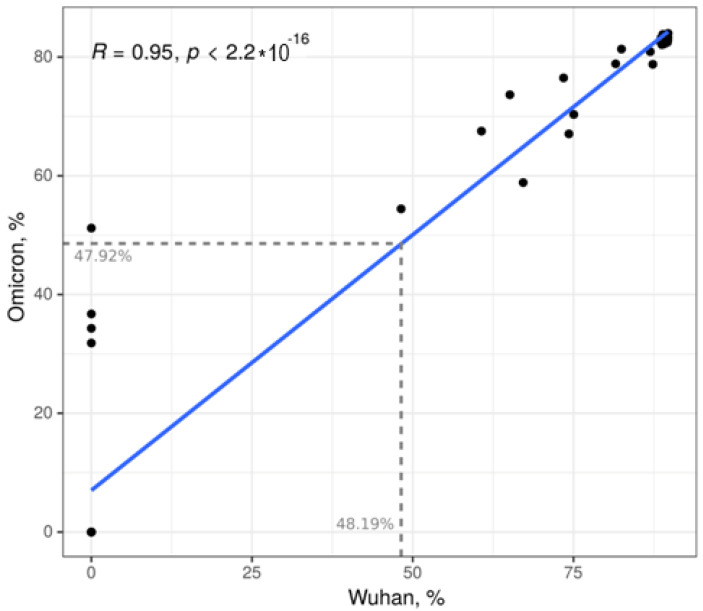
Correlation analysis of neutralizing antibodies to Omicron and Wuhan (based on Spearman’s rank correlation coefficient).

**Figure 3 ijms-24-10181-f003:**
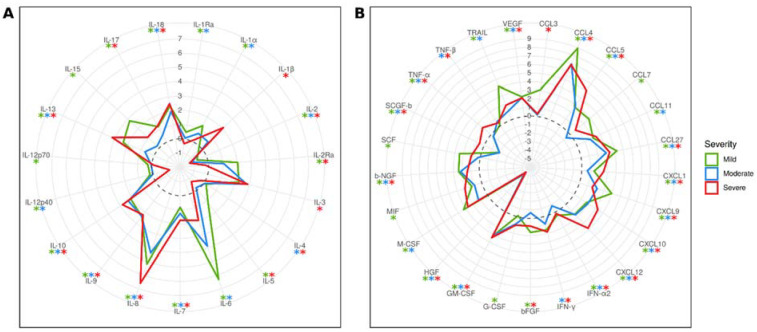
Differences in serum cytokines levels in different severity COVID-19 compared to controls. Data are presented as log_2_(mean (COVID-19)/mean (Control)). (**A**)—interleukins. (**B**)—other cytokines. *—*p* < 0.05, Kruskal–Wallis test with Benjamini–Hochberg adjustment for multiple comparisons.

**Figure 4 ijms-24-10181-f004:**
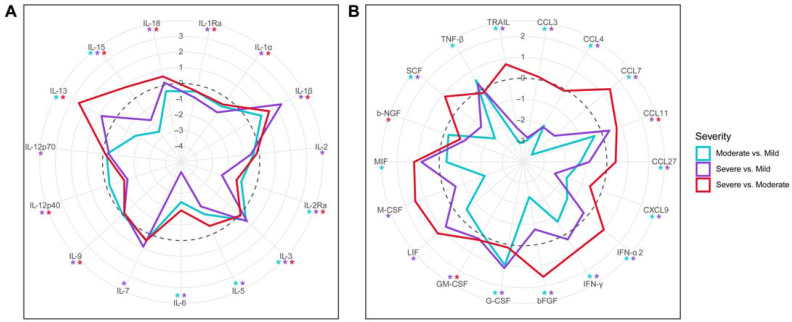
Differences in serum cytokines levels in different severity COVID-19. Data are presented as log_2_(mean (COVID-19)/mean (COVID-19)). (**A**)—interleukins. (**B**)—other cytokines. *—*p* < 0.05, Kruskal–Wallis test with Benjamini–Hochberg adjustment for multiple comparisons.

**Figure 5 ijms-24-10181-f005:**
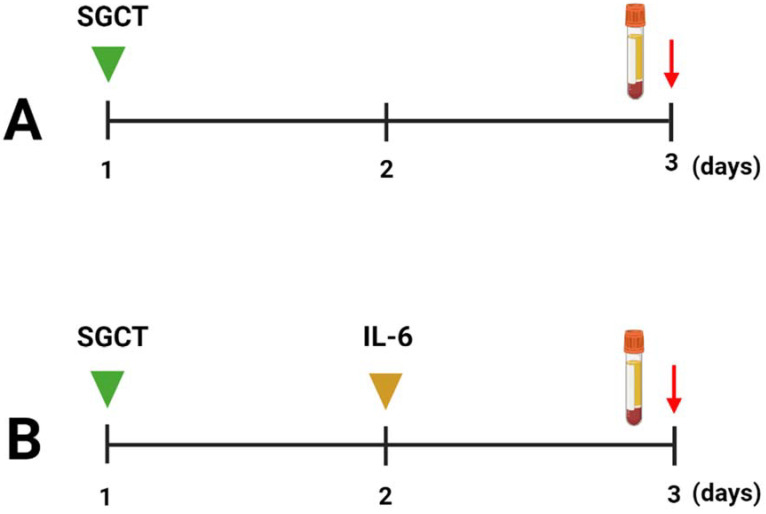
Schematic presentation of (**A**) SGCT and (**B**) olokizumab treatment of COVID-19. Green arrow—initiation of SGCT; Brown arrow—single dose olokizumab treatment; Red arrow—sample collection.

**Table 1 ijms-24-10181-t001:** COVID-19 clinical laboratory data.

	Control (*n* = 26)	COVID-19			*p* Value					
		Mild (*n* = 24)	Moderate (*n* = 30)	Severe (*n* = 27)	1	2	3	4	5	6
Sex (m/f)	11/15	7/17	15/15	10/17	0.39 §	0.60 §	0.78 §	0.17 §	0.77 §	0.42 §
Age	62.8 ± 13.6	64.3 ± 17.3	65.7 ± 14.0	70.8 ± 12.4	0.40	0.60	0.14	0.81	0.43	0.43
Leu, 10^9^	5.3 ± 1.2	5.2 ± 2.4	6.2 ± 2.9	7.9 ± 4.5	0.46	0.42	0.11	0.19	0.03	0.33
Monocyte, %	3.5 ± 0.8	10.2 ± 4.6	8.8 ± 3.5	5.6 ± 2.5	<0.01	<0.01	0.02	0.37	<0.01	<0.01
Monocyte, 10^9^	0.19 ± 0.06	0.48 ± 0.21	0.48 ± 0.19	0.41 ± 0.28	<0.01	<0.01	<0.01	0.87	0.15	0.11
Lyphocyte, %	26.5 ± 4.1	31.0 ± 13.7	27.3 ± 15.8	15.0 ± 11.0	0.45	0.96	<0.01	0.51	<0.01	<0.01
Lyphocyte, 10^9^	1.42 ± 0.47	1.46 ± 0.72	1.43 ± 0.76	0.89 ± 0.60	0.70	0.58	<0.01	0.78	<0.01	<0.01
Platelets, 10^9^	210.1 ± 10.0	201.3 ± 73.6	212.9 ± 81.3	218.4 ± 86.8	0.46	0.93	0.69	0.49	0.58	0.96
Hb, g/L	134.8 ± 12.6	131.7 ± 15.8	129.1 ± 20.5	129.7 ± 16.7	0.78	1.00	1.00	1.00	0.96	0.91
CRP, mg/L	1.2 ± 0.5	21.4 ± 34.9	56.1 ± 79.4	99.4 ± 68.4	<0.01	<0.01	<0.01	0.04	<0.01	0.049
ALT	22.2 ± 4.1	26.3 ± 17.4	31.3 ± 20.4	36.1 ± 22.0	0.96	0.45	0.22	0.58	0.42	0.53
AST	29.3 ± 3.7	35.6 ± 25.5	37.3 ± 22.4	51.4 ± 34.4	0.89	0.46	0.01	0.48	0.02	0.09
TT, sec	13.9 ± 1.5	14.7 ± 9.0	17.4 ± 5.1	19.2 ± 11.9	0.04	<0.01	<0.01	0.17	0.08	0.56
Fbrinogen, mg/L	1.5 ± 0.2	21.8 ± 77.3	4.4 ± 1.3	5.3 ± 2.3	<0.01	<0.01	<0.01	0.92	0.50	0.50
CT, % lung damage	-	4.4 ± 6.9	20.0 ± 13.4	47.7 ± 25.9	-	-	-	<0.01	<0.01	<0.01
Omicron neutralizing antibodies	0.0 ± 0.0	19.5 ± 34.9	43.6 ± 38.0	12.1 ± 27.4	0.06	<0.01	0.19	0.02	0.48	<0.01
Wuhan neutralizing antibodies	0.0 ± 0.0	48.7 ± 43.7	46.2 ± 40.9	26.6 ± 42.8	<0.01	<0.01	0.15	0.76	0.27	0.22
Anti-SARS-CoV-2 antibody	0.051 ± 0.069	0.959 ± 1.433	1.794 ± 1.519	0.802 ± 1.254	0.01	<0.01	0.01	0.03	0.65	0.01

1—Control vs. Mild COVID-19; 2—Control vs. Moderate COVID-19; 3—Control vs. Severe COVID-19; 4—Mild COVID-19 vs. Moderate COVID-19; 5—Mild COVID-19 vs. Severe COVID-19; 6—Moderate COVID-19 vs. Severe COVID-19. *p* values were calculated using Kruskal–Wallis test with BH correction, §—Exact Fisher test with BH correction.

**Table 2 ijms-24-10181-t002:** Frequency of anti-Omicron neutralizing antibodies in COVID-19 patients.

	Positive	Negative	*p* Value
Mild COVID-19	6 (25%)	18 (75%)	Mild vs. Moderate: 0.02
Moderate COVID-19	18 (60%)	12 (40%)	Mild vs. Severe: 0.74
Severe COVID-19	5 (18.5%)	22 (81.5%)	Moderate vs. Severe: 0.008

*p* values were calculated using Kruskal–Wallis test with BH correction.

## Data Availability

All data are presented in the text.

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
