# Peer review of "Neutralizing Antibodies in COVID-19 Serum from Tatarstan, Russia"

_ijms, 2023, doi:10.3390/ijms241210181_

Round 1

Reviewer 1 Report

I reviewed the manuscript ijms-2413122 by Hamza S. et al. The authors evaluated the presence of neutralizing antibodies against Wuhan and Omicron variants of SARS-CoV-2 and the cytokine pattern in plasma samples from 81 patients with COVID-19. They demonstrated that patients with moderate disease exhibited an increased number of neutralizing abs compared to both pts with mild and severe COVID-19. Furthermore, the abs were cross-reactive against Wuhan and omicron variants. Interestingly, the authors observed that pts with severe COVID-19 had a Th-17-related cytokine profile. The manuscript is generally well-written. However some defects are present. The most important ones are the following:

1) The authors should include in table 1 both the percentages and absolute values of white blood cells.

2) They should include the time points of blood sampling of all 81 pts and provide detailed description of specific ant-SARS-CoV-2 therapies for all groups (e.g. corticosteroids, baricitinb, tocilizumab etc). They should also state whether blood was obtained before or after specific treatment administration.

3) The authors describe that they did not observe differences in neutralizing antibodies between fatal and non-fatal cases. However they demonstrate that death rate in pts with severe COVID-19 was 51.9% and in pts with moderate disease was 3.33%. Since a statistically significant difference in neutralizing abs has been observed between overall pts with moderate disease compared to those with severe disease, the authors should provide an explanation why this difference was not observed between fatal and non-fatal cases.

4) A recent publication by Pappas AG, et al. (Kinetics of Immune Subsets in COVID-19 Patients Treated with Corticosteroids. Viruses. 2022 Dec 24;15(1):51. doi: 10.3390/v15010051.) demonstrated that pts with COVID-19-related respiratory failure who survived and did not require mechanical ventilation exhibited a relatively higher increase in the number of B-lymphocytes over time compared to the patients who intubated or died and the patients without respiratory failure. The authors should comment in the discussion section their findings under this prism.  

Author Response

I reviewed the manuscript ijms-2413122 by Hamza S. et al. The authors evaluated the presence of neutralizing antibodies against Wuhan and Omicron variants of SARS-CoV-2 and the cytokine pattern in plasma samples from 81 patients with COVID-19. They demonstrated that patients with moderate disease exhibited an increased number of neutralizing abs compared to both pts with mild and severe COVID-19. Furthermore, the abs were cross-reactive against Wuhan and omicron variants. Interestingly, the authors observed that pts with severe COVID-19 had a Th-17-related cytokine profile. The manuscript is generally well-written. However some defects are present. The most important ones are the following:

  • The authors should include in table 1 both the percentages and absolute values of white blood cells.

Agree: changes were made. The absolute values were included.

2) They should include the time points of blood sampling of all 81 pts and provide detailed description of specific ant-SARS-CoV-2 therapies for all groups (e.g. corticosteroids, baricitinb, tocilizumab etc). They should also state whether blood was obtained before or after specific treatment administration.

Agree: the detailed description of anti-SARS-CoV-2 therapies and time when blood samples were collected is stated in lines: 108-118:

2.2. COVID-19 treatment

All patients received standard glucocorticoid therapy (SGCT) as part of their treatment. However, for patients who did not show signs of clinical improvement (such as increased body temperature, lower PaO2 and PaO2/FiO2, progressive respiratory insufficiency, etc.), an additional intervention was administered. These patients were given olokizumab, an IL-6 binding monoclonal antibody [32], 24 hours after initiating SGCT. The SGCT regimen began upon admission and involved the intravenous administration of methylprednisolone (60 mg) every 6 hours for a duration of 4 days. The dosage was subsequently reduced by 20-25% every 1-2 days, with a 50% decrease occurring after 2 days. Twenty-four hours after initiating SGCT, patients received a single dose of olokizumab (256 mg) via intravenous injection. A schematic presentation of the COVID-19 treatment approach is depicted in Figure 1.

Figure 1. Schematic presentation of SGCT and olokizumab treatment of COVID-19.

Green arrow – initiation of SGCT; Brown arrow – single dose olokizumab treatment; Red arrow – sample collection.

3) The authors describe that they did not observe differences in neutralizing antibodies between fatal and non-fatal cases. However they demonstrate that death rate in pts with severe COVID-19 was 51.9% and in pts with moderate disease was 3.33%. Since a statistically significant difference in neutralizing abs has been observed between overall pts with moderate disease compared to those with severe disease, the authors should provide an explanation why this difference was not observed between fatal and non-fatal cases.

Answer: When analyzing the data of both fatal and non-fatal patients, we included all 81 patients, encompassing individuals with mild, moderate, and severe COVID-19. We believe that the inclusion of mild cases, which did not result in fatalities, may have contributed to the lack of statistically significant differences between fatal and non-fatal COVID-19 cases. However, when comparing the moderate group with either the mild or severe groups, we did observe a statistically significant difference. This was due to the higher frequency of neutralizing antibodies in the moderate group compared to the mild and severe groups.

It is important to note that almost all fatal COVID-19 cases were classified as severe forms of the disease. Among patients with severe COVID-19, we found that 1 out of 14 non-fatal cases (7%) and 4 out of 15 fatal cases (26%) had detectable neutralizing antibodies. While it appears that there were more patients with neutralizing antibodies in the fatal COVID-19 group compared to the non-fatal group, this difference was not statistically significant (p=0.33, Exact Fisher test). These findings suggest that neutralizing antibodies may have a limited impact on the severity of the disease. It is possible that neutralizing antibodies play a more significant role in protecting against initial infection, while their contribution to the severity of the disease, which is primarily driven by a severe inflammatory response to SARS-CoV-2 infection, may be limited.

4) A recent publication by Pappas AG, et al. (Kinetics of Immune Subsets in COVID-19 Patients Treated with Corticosteroids. Viruses. 2022 Dec 24;15(1):51. doi: 10.3390/v15010051.) demonstrated that pts with COVID-19-related respiratory failure who survived and did not require mechanical ventilation exhibited a relatively higher increase in the number of B-lymphocytes over time compared to the patients who intubated or died and the patients without respiratory failure. The authors should comment in the discussion section their findings under this prism.  

Agree: Data published by Pappas et al is included into the discussion section. Lines 317-321

Additionally, the lower prevalence of anti-SARS-CoV-2 antibodies among severe COVID-19 patients could be attributed to the observed decrease in CD19+ lymphocyte numbers in this particular patient group [43]. Papas et al. conducted a study that revealed significantly lower B cell counts in severe COVID-19 patients, and these counts did not recover throughout the course of the disease [43].

Reviewer 2 Report

This study suffers from essential issues regarding the study design and the derived conclusions:

1. The methods are not explained in detail and it’s hard to follow the employed strategies. The methods should be explain in more details.

2. It is not clear how the authors determined if the participants do not experience prior SARSCOV-2 infection. It should be determined that the participants experienced prior SARSCOV-2 infection and how many times. The neutralizing antibodies could be the result of prior SARSCOV-2 infections and not the ongoing infection or the vaccination.

4. The number of enrolled individuals could be higher to make a more informed conclusions

5. The effect of ethnicity and geographical differences does not taken in to account at all to derive the conclusions.

6. Various studies have already done similar studies which reduces the novelty of this study to a great deal

minor revision is required

Author Response

This study suffers from essential issues regarding the study design and the derived conclusions:

  1. The methods are not explained in detail and it’s hard to follow the employed strategies. The methods should be explain in more details.

Agree: changes were made in the description of methods to provide more explanation.

  1. It is not clear how the authors determined if the participants do not experience prior SARSCOV-2 infection. It should be determined that the participants experienced prior SARSCOV-2 infection and how many times. The neutralizing antibodies could be the result of prior SARSCOV-2 infections and not the ongoing infection or the vaccination.

Answer: We did not collect data on prior COVID-19 infections in our patient cohorts, and we acknowledge that previous COVID-19 infection can impact the antibody immune response. It is important to consider the influence of prior infections on the results.

However, there are several reasons why we believe our data is still relevant to the current state of COVID-19. Studies have indicated a high risk of reinfection between 6-12 months after recovery from COVID-19 (https://www.ncbi.nlm.nih.gov/pmc/articles/PMC8506664/ , https://www.ncbi.nlm.nih.gov/pmc/articles/PMC7989568/ , https://www.ncbi.nlm.nih.gov/pmc/articles/PMC7988582/ ) (reviewed in https://www.ncbi.nlm.nih.gov/pmc/articles/PMC8824301/ ). Although still subject to debate, it is suggested that the immune response, including the humoral response, plays a role in providing protection against re-infection. Therefore, experiencing a COVID-19 re-infection could indicate that the protective capacity of the immune response has not been substantially reduced. However, it is important to consider that re-infection could also be a result of infection with a different strain of SARS-CoV-2, where the previous immune response may have limited effectiveness.

Additionally, it is worth noting that COVID-19 infections can occur in asymptomatic or mild forms, and individuals may not be aware that they have been infected with SARS-CoV-2. This can further complicate the understanding and assessment of prior infection history.

Studies have shown that the previous SARS-CoV-2 infection could lead to mild form of the disease (https://pubmed.ncbi.nlm.nih.gov/33853339/  , https://www.ncbi.nlm.nih.gov/pmc/articles/PMC8114888/ , https://www.ncbi.nlm.nih.gov/pmc/articles/PMC8535385/ ). However, in our study as well as published by others (https://www.ncbi.nlm.nih.gov/pmc/articles/PMC7408950/ , https://academic.oup.com/ofid/article/9/3/ofac055/6541150 ) we observed a lower prevalence of neutralizing antibodies in mild COVID-19 patients compared to severe COVID-19 patients. This suggests that prior infection may have had a limited contribution to the development of neutralizing antibodies in the patient group included in our study.

  1. The number of enrolled individuals could be higher to make a more informed conclusions

Answer: the number of COVID-19 patients enrolled in this study is 81, which is close to the numbers used in other studies listed here: 48 patients (https://www.thelancet.com/journals/ebiom/article/PIIS2352-3964(22)00209-2/fulltext ), 63 patients (https://www.microbiologyresearch.org/content/journal/acmi/10.1099/acmi.0.000257?crawler=true ), 89 patients (https://journals.lww.com/ijmr/Fulltext/2020/52010/Neutralizing_antibody_responses_to_SARS_CoV_2_in.12.aspx ), 111 patients (https://www.sciencedirect.com/science/article/pii/S1198743X21007205 ), 100 patients (https://www.mdpi.com/2076-393X/10/8/1312 ) and 111 patients (https://www.ncbi.nlm.nih.gov/pmc/articles/PMC7442695/ ).

  1. The effect of ethnicity and geographical differences does not taken in to account at all to derive the conclusions.

Agree: In response to the reviewer's comment, we have completed the analysis of neutralizing antibody detection in patients with different ethnicities. The data is presented in Supplemental Table 3 and 4, and the description can be found in lines 240-241 of the manuscript.

According to Census 2021 data (source: https://en.wikipedia.org/wiki/Tatarstan), the Republic of Tatarstan has a population comprising 53% Tatar and 40% Russian ethnic groups. It is worth noting that Tatarstan has had a history of mixed ethnicity for centuries, dating back to the 17th century. Current studies indicate that the population is in a state of harmony, with numerous inter-ethnic marriages being registered (source: https://www.ersj.eu/dmdocuments/02_06.12.17.pdf). Therefore, determining specific ethnic groups can be challenging, as an individual may be registered as Tatar by ethnicity but have Russian ancestors (such as grandparents or parents). Genetic testing for specific genetic markers could potentially aid in distinguishing between ethnic groups.

Our analysis revealed that the ethnicity of patients did not contribute to the severity of the disease, as there was no significant difference in the frequency of mild, moderate, and severe forms of COVID-19 among patients with different ethnic backgrounds (Supplemental Table 3). Additionally, there was no significant difference in the frequency of neutralizing antibodies among patients with different ethnicities, regardless of whether they had mild, moderate, or severe forms of the disease (Supplemental Table 4).

  1. Various studies have already done similar studies which reduces the novelty of this study to a great deal

Agreed/Answered: Our current understanding of neutralizing antibody activation in COVID-19 is primarily based on data collected from studies conducted in Europe, the USA, and China. However, there is a scarcity of data on neutralizing antibody activation in COVID-19 patients from Russia. In this manuscript, we present a novel study that examines the activation of neutralizing antibodies in patients diagnosed with COVID-19 in the Republic of Tatarstan.

This study is unique as it provides a comprehensive analysis of neutralizing antibodies in patients with varying severities of COVID-19, different treatment regimens, diverse sexes, and ethnic backgrounds. Our research also investigates the cross-reactivity of neutralizing antibodies between the Wuhan and Omicron variants of SARS-CoV-2. By exploring these aspects, we aim to contribute to the existing knowledge and understanding of neutralizing antibody responses in the context of COVID-19, particularly in the specific region of Tatarstan, Russia.

To address the Reviewer’s comment, we added the statement to clarify the significance of this study Lines: 76-83

COVID-19 was first diagnosed in the Republic of Tatarstan, Russia in March 2020 [28]. Since then, several variants of the SARS-CoV-2 virus, such as Delta and Omicron, have been isolated from COVID-19 patients in Russia [29]. While there is limited data available on the antibody response to SARS-CoV-2 [30], even less is known about the production of neutralizing antibodies in COVID-19 patients in Russia [31]. To address this knowledge gap, our study aimed to analyze the humoral immune response in mild, moderate, and severe COVID-19 patients from the Republic of Tatarstan, Russia, with a specific focus on studying neutralizing antibodies. 

Reviewer 3 Report

The manuscript “Neutralizing antibodies in COVID-19 serum from Tatarstan, Russia” by Shaima Hamza and co-workers analyzed the neutralizing antibodies in patients with mild, moderate, and severe COVID-19 and cross-reactivity with the RBD of spike protein from Wuhan and Omicron strains. They also compared the cytokines change among mild, moderate, and severe COVID-19  patients. The results yielded from this study will provide us with more information about the relationship between neutralizing antibodies and disease development. I have several questions about this manuscript:

Major concerns

1, The sample sizes are small, which will compromise the credibility of the results from statistical analysis.

2, The development of neutralizing antibodies is closely related to the time of SARS-CoV-2  infection. Please include the collection time information (xx days after infection or symptoms ) of serums and analyze the effects on neutralizing antibodies.

3, SARS-CoV-2 Omicron variant was prevalent in Russia at the time of the serum collection. However, it still cannot rule out other SARS-CoV-2 variants(for example, SASR-CoV-2 delta strain) circulating at the same time. So authors have to figure out which strains of SARS-CoV-2 are responsible for the infection of the patients whose samples were collected for this study. This information is important because it directly associates with the cross-reactivity of neutralizing antibodies and even the conclusion(s).

Minor concerns

1.     The neutralizing assay utilized in this study is not a standard neutralizing method. Please clarify how much the neutralizing assay in this study can reproduce the results from the standard Neutralization Assay Plaque Neutralization (PRNT).

2.     Suggest to simplify the discussion part, especially for the paragraph (line316 to 332). There is a lot of background information but little information closely related to the discussion of the results from this study. The essential background should be put in the introduction part.

Lnie 27: ”SARS-CoV-2  pandemic ”should be  “COVID-19 pandemic”

Line 28: Please revise this sentence since the WHO declared an end to the COVID-19 pandemic.

Line 29: The resulting COVID-19?

Line39-40: “which binds to the angiotensin-converting enzyme 2 receptor on the host cell membrane” suggest to change to “which mediates SARS-CoV-2 cell entry by binding to the angiotensin-converting enzyme 2 receptor on the host cell membrane.”

Line 219 “All anti-Omicron neutralizing antibody positive mild and severe COVID-19 serum samples”?

line219-222: It is difficult to understand those sentences. Please rewrite them to make them explicit.

The quality of the English language is good, but some places need to be revised to make it more clear.

Author Response

Rev 3

he manuscript “Neutralizing antibodies in COVID-19 serum from Tatarstan, Russia” by Shaima Hamza and co-workers analyzed the neutralizing antibodies in patients with mild, moderate, and severe COVID-19 and cross-reactivity with the RBD of spike protein from Wuhan and Omicron strains. They also compared the cytokines change among mild, moderate, and severe COVID-19  patients. The results yielded from this study will provide us with more information about the relationship between neutralizing antibodies and disease development. I have several questions about this manuscript:

Major concerns

1, The sample sizes are small, which will compromise the credibility of the results from statistical analysis.

 Answer: the number of COVID-19 patients enrolled in this study is 81, which is close to the numbers used in other studies listed here: 48 patients (https://www.thelancet.com/journals/ebiom/article/PIIS2352-3964(22)00209-2/fulltext ), 63 patients (https://www.microbiologyresearch.org/content/journal/acmi/10.1099/acmi.0.000257?crawler=true ), 89 patients (https://journals.lww.com/ijmr/Fulltext/2020/52010/Neutralizing_antibody_responses_to_SARS_CoV_2_in.12.aspx ), 111 patients (https://www.sciencedirect.com/science/article/pii/S1198743X21007205 ), 100 patients (https://www.mdpi.com/2076-393X/10/8/1312 ) and 111 patients (https://www.ncbi.nlm.nih.gov/pmc/articles/PMC7442695/ ).

2, The development of neutralizing antibodies is closely related to the time of SARS-CoV-2  infection. Please include the collection time information (xx days after infection or symptoms ) of serums and analyze the effects on neutralizing antibodies.

 Agree: the day of samples collection was included in lines: 92-93. Samples were collected on third day post admission to the hospital.

3, SARS-CoV-2 Omicron variant was prevalent in Russia at the time of the serum collection. However, it still cannot rule out other SARS-CoV-2 variants(for example, SASR-CoV-2 delta strain) circulating at the same time. So authors have to figure out which strains of SARS-CoV-2 are responsible for the infection of the patients whose samples were collected for this study. This information is important because it directly associates with the cross-reactivity of neutralizing antibodies and even the conclusion(s).

  Answer: At the time of admission, nasopharyngeal swabs were collected from the patients and tested for SARS-CoV-2 RNA using PCR by the hospital laboratory personnel. The PCR test conducted in the clinical laboratory was not specific to a particular SARS-CoV-2 virus strain. The healthcare providers explained that the knowledge of the specific strain of the virus did not impact the treatment plan, which primarily focused on reducing inflammation and supporting vital functions.

PCR-positive samples were subsequently sent to the center for infectious control (RosPotrebNadzor), where virus-specific PCR tests were performed to monitor changes in circulating SARS-CoV-2 virus strains.

We acknowledge that some COVID-19 cases in December 2021 could have been caused by the delta variant of SARS-CoV-2. However, according to RosPotrebNadzor data, by February 2022, all COVID-19 cases were caused by the omicron variant, and no cases caused by the delta variant were documented (source: https://www.statista.com/statistics/1286435/sars-cov-2-omicron-variant-share-russia/). The omicron variant was first detected in November 2021 and rapidly spread, resulting in approximately 50% It is important to note that during January 2022, both the delta variant and the omicron variant of SARS-CoV-2 were circulating in Russia simultaneously [23]. How-ever, our study specifically focused on the detection of omicron SARS-CoV-2 neutralizing antibodies in the COVID-19 patients included in this study. It is possible that some patients may have been infected with the delta variant and developed non-cross-reacting neutralizing antibodies. Nevertheless, the key finding of our study is that neutralizing antibodies against the omicron variant of SARS-CoV-2 were produced in COVID-19 patients. These antibodies are expected to provide protection against reinfection with the omicron variant, which became the dominant strain in Russia after February 2022.  In response to the reviewer's comment, we re-evaluated the dates of patient serum collection. After careful analysis, we have corrected the dates from December 2021 - March 2022 to January 2022 - March 2022. However, we acknowledge that there is a possibility of some delta variant of SARS-CoV-2 circulating in the Republic of Tatarstan.

To acknowledge the valid point of the reviewer, we included the statement about omicron variant prevalence in Russia during January - March 2022. We also added statement about possibility of some patients having anti-delta SARS-CoV-2 variant neutralizing antibodies (lines 304-312)

It should be noted that during January 2022, there were delta variant circulating in Russia simultaneously with addition to omicron variant of SARS-CoV-2 [23]. Our study was limited to detection of omicron SARS-CoV-2 neutralizing antibodies in COVID-19 patients included in this study. There is a possibility that some patients were infected with delta variant and developed non-cross reacting neutralizing antibodies. Still, the important conclusion is that neutralizing anti-omicron SARS-CoV-2 antibodies were produced in COVID-19 patients. These antibodies would provide a protection against the re-infection with omicron SARS-CoV-2 variant, which became a dominant after February 2022 in Russia.

Minor concerns

  1. The neutralizing assay utilized in this study is not a standard neutralizing method. Please clarify how much the neutralizing assay in this study can reproduce the results from the standard Neutralization Assay Plaque Neutralization (PRNT).

Answer:

The protocol description provided by the manufacturer included information on the comparison of the GenScript cPass SARS-CoV-2 neutralizing test results with the Plaque Reduction Neutralization Assay (PRNT). The manufacturer stated that to validate the clinical performance of the GenScript cPass SARS-CoV-2 Neutralization Antibody Detection Kit, they utilized the comparator PRNT, which employed the SARS-CoV-2 virus (WA01/2020 isolate).

In the clinical agreement study, a total of 114 samples were retrospectively collected from individuals who tested positive or negative for SARS-CoV-2 by RT-PCR (26 PRNT positive and 88 PRNT negative samples). Both the cPass™ SARS-CoV-2 Neutralization Antibody Detection Kit and the PRNT comparator (PRNT50 and PRNT90) were used for these samples. The study cohort consisted of samples from normal healthy individuals (n=88) as well as samples from RT-PCR confirmed SARS-CoV-2 positive patients (n=26).

The GenScript cPass SARS-CoV-2 Neutralization Antibody Detection Kit results were compared to the Plaque Reduction Neutralization Test performed according to WHO guidelines. The protocol included tables showing the Positive and Negative Percent Agreement between the PRNT50 or PRNT90 and the cPass SARSCoV-2 Neutralization Antibody Detection Kit results.

The manufacturer states that the GenScript cPass SARS-CoV-2 neutralizing test has 100% positive agreement with PRNT50 (95% CI 87.1-100.0%) and 100% positive agreement with PRNT90 (95% CI 87.1-100.0%). This information is added to the text lines: 159-161

This test was clinically validated by the manufacturer using the Plaque Reduction Neutralization Assay (PRNT) as a standard assay to detect neutralizing antibodies (https://www.genscript.com/product/documents?cat_no=L00847-A&catalogtype=Document-PROTOCOL). The manufacturer states that the GenScript cPass SARS-CoV-2 neutralizing test has 100% positive agreement with PRNT50 (95% CI 87.1-100.0%) and 100% positive agreement with PRNT90 (95% CI 87.1-100.0%).

  1. Suggest to simplify the discussion part, especially for the paragraph (line316 to 332). There is a lot of background information but little information closely related to the discussion of the results from this study. The essential background should be put in the introduction part.

 Agree: changes were made to simplify the paragraph (lines 356-364) by substantially reducing its content.

Another notable finding was the elevated levels of IL-1β and IL-17 in the serum of severe COVID-19 patients compared to the control group. IL-1β is a pleiotropic cytokine that is produced by activated inflammasomes [52]. Boraschi summarizes the role of IL-1β in the pathogenesis of the disease as initiating and amplifying the inflammatory response, as well as recruiting and activating leukocytes [53]. IL-1β can also support T cell priming [54] and enhance the release of IFN-γ and IL-17 by CD4+ T cells [55-56]. On the other hand, IL-17 is primarily produced by Th17 lymphocytes and is implicated in the pathogenesis of immunopathology [57]. IL-17 can work synergistically with IL-1β and TNF-α to enhance inflammation [58]. Hence, the role of IL-17 in COVID-19 has been recognized as an "inflammation amplifier" [59].

Lnie 27: ”SARS-CoV-2  pandemic ”should be  “COVID-19 pandemic”

Agree: changes were made

Line 28: Please revise this sentence since the WHO declared an end to the COVID-19 pandemic.

Agree: changes were made

Line 29: The resulting COVID-19?

Agree: changes were made (line 28). Revised sentences is; “COVID-19 is a disease characterized by symptoms such as fever, cough, sore throat, shortness of breath, loss of sense of smell and taste”.

Line39-40: “which binds to the angiotensin-converting enzyme 2 receptor on the host cell membrane” suggest to change to “which mediates SARS-CoV-2 cell entry by binding to the angiotensin-converting enzyme 2 receptor on the host cell membrane.”

Agree: changes were made (lines 38-40). Revised sentence is: “The majority of these antibodies are directed against the spike (S) protein [6-7], which mediates SARS-CoV-2 cell entry by binding to the angiotensin-converting enzyme 2 receptor on the host cell membrane [8].”

Line 219 “All anti-Omicron neutralizing antibody positive mild and severe COVID-19 serum samples”?

Agree: changes were made in sentence (lines 219-220): Only anti-Omicron neutralizing antibody positive mild and severe COVID-19 serum samples were selected for this study.

line219-222: It is difficult to understand those sentences. Please rewrite them to make them explicit.

Agree: sentences were modified for clarity: lines 277-278:

Additionally, we included 10 and 11 serum samples from mild and severe COVID-19, respectively. Only serum samples positive for anti-Omicron neutralizing from mild and severe COVID-19 patients were selected for this study.

Round 2

Reviewer 2 Report

the authors have answered the comments adequately.

minor revisions would be beneficial.

Reviewer 3 Report

There are no more comments

The language is fine